# Assessing Spurious Correlations in Big Search Data

**Jesse T. Richman** [1,*] and **Ryan J. Roberts** [2]

1 Department of Political Science and Geography, Old Dominion University, BAL 7000, Norfolk, VA 23529, USA
2 Department of Public Service, Gardner-Webb University, Boiling Springs, NC 28017, USA
* Correspondence: jrichman@odu.edu; Tel.: +1-757-683-3853

**Abstract:** Big search data offers the opportunity to identify new and potentially real-time measures and predictors of important political, geographic, social, cultural, economic, and epidemiological phenomena, measures that might serve an important role as leading indicators in forecasts and nowcasts. However, it also presents vast new risks that scientists or the public will identify meaningless and totally spurious 'relationships' between variables. This study is the first to quantify that risk in the context of search data. We find that spurious correlations arise at exceptionally high frequencies among probability distributions examined for random variables based upon gamma (1, 1) and Gaussian random walk distributions. Quantifying these spurious correlations and their likely magnitude for various distributions has value for several reasons. First, analysts can make progress toward accurate inference. Second, they can avoid unwarranted credulity. Third, they can demand appropriate disclosure from the study authors.

**Keywords:** spurious correlation; Bonferroni; big data; big search data; Google Correlate; Google Trends; search data





## 1. Introduction

"Fat big data" [1] is data with so many potential variables that the number of variables exceeds the number of observations in the dataset. Such data often offers researchers the opportunity to examine huge numbers of potential correlates of the phenomenon of interest. That is certainly true of the type of big data this study focuses on – time series and cross-sectional search frequency data, henceforth big search data. Search data has been used in a variety of studies across many disciplines for nearly two decades [2–16] to predict or measure a phenomenon of interest. Articles promoting its use in economics [4] and public health [15] had more than 3000 and more than 5000 citations in Google scholar, respectively, by early 2023. The potential is enormous. For instance, Rashad [14] found that "Google data can significantly improve tourism forecasting and serves as a leading indicator of tourism demand". The challenge with using big search data is that it can readily mislead if mishandled—serving up unreliable or spurious correlations instead of actual insights [16–21]. One study recently suggested [17] that big data risks providing "building blocks for a new type of ignorance that is not an absence of knowledge, but the presence of something masquerading as it".

All big data raises a number of interrelated challenges for inference. Meng [22] emphasizes that in the context of estimating a single population parameter such as a mean or vote share, analysts must take into account a "data quality measure, ρ". According to Meng, the data quality measure is part and parcel of making accurate estimates, along with population and sample size and the difficulty or standard deviation of the data. The analysis reported below highlights the need for a parallel analysis extending work [1,18–20,23–26] involving alleged correlations between variables. This work could include more systematically examining correlations with variables in a wide range of datasets, not just the search data examined here. Ideally, as Meng suggests for the single-variable case, one could use appropriate adjustments or priors to help structure interpretation in particular contexts.

The main focus of this study is an illustrative analysis of spurious correlation in big search data relevant to estimating such priors.

Spurious correlations are likely to be frequent in most big data (including big search data). The key criterion is that the data have many potential variables or measures [1,20]. As Vigen [21] and Calude and Longo [18] emphasize, it is important to examine the topic of spurious correlations in big data because big data creates vast new opportunities to find and be misled by completely spurious correlations. Apparent relationships between variables have no causal connection whatsoever. The basic statistical logic is well known. The availability of millions of potential variables dramatically raises the odds of high correlations backed by no underlying causal relationship. Erroneous forecasts based upon spurious correlations might be called Bonferroni's revenge [23–25]. For instance, the Bonferroni correction with $\alpha = 0.05$ and ten million hypotheses tested is $0.05/10{,}000{,}000 = 0.000000005$.

This study operates in a middle plane theoretically and empirically between Vigen [21] and Calude and Longo [18]. Calude and Longo prove that with sufficiently fat data, spurious correlations of any magnitude can be identified, but they do not examine actual spurious correlations with real data, nor do they demonstrate how large the data need to be to find correlations of a particular magnitude. Vigen [21] collected a large number of short annual time series and mined this dataset for correlations between variables (e.g., between money spent on pets and alcohol purchased in liquor stores from 2000 to 2009: p.87, correlation 99.4%). However, Vigen made no effort to quantify the likelihood of observing spurious correlations or the overall probability of large spurious correlations. Vigen initially published a website of these data cartoons https://tylervigen.com/spurious-correlations (accessed on 24 February 2023) and then followed up with a book. While delightfully amusing, Vigen's data cartoons do not provide much guidance concerning the magnitude of the spurious correlation problem in any domain.

As noted earlier, the focus of our study is what might be thought of as a case study of a particular kind of big data—big search data. This data measures the temporal and spatial frequency with which internet search engine users search for particular words and phrases. The main ongoing portal to access Google search frequency data is the Google Trends tool https://trends.google.com/trends/ (accessed on 24 February 2023), which allows users to see and download data on the frequency with which search terms have been used (in particular geographic areas and over particular time frames). Google Correlate [27] is a now-discontinued tool that we used to collect the data reported in this study (shortly before it was discontinued). Google Correlate allowed users to upload a cross-sectional or time series variable. The tool then suggested search terms that had search frequency data with high correlations with the uploaded variable. To collect the data used in this article, as described below, we used Google Correlate to identify search results that were highly correlated with the random-number-generated variables.

This paper quantifies the risk of spurious correlations to raise awareness and guide response. Quantifying the risks of spurious correlation and identifying paths to minimize the risks should help scholars better use big search data for forecasting and other purposes. This study is the first to quantify the problem of spurious correlation in big search data–the magnitude and prevalence of the spurious correlations that regularly arise for various types of statistical distributions. This extends Vigen [21] by making the exploration of spurious correlations more systematic. It extends Calude and Longo [18] by systematically examining the prevalence of spurious correlation with real data.

Quantifying the magnitude of the spurious correlation challenge is critical because it can point the path toward potential solutions. That very large spurious correlations can arise at alarmingly high frequencies in big search data should not be a surprise, perhaps, given foundational work in statistics on repeated tests [21,23–25]. Nonetheless, there is value to empirical examination. For example, one might not know the risk of spurious correlation is lower if the variable one is attempting to predict is drawn from a uniform distribution compared to a gamma distribution, as demonstrated here.

Of course, this is only a first step. Solving the spurious correlation issue is difficult. L'Heureux et al. [23] review the range of approaches to addressing inference in big data and find no approaches that fully address the challenge of spurious correlation. Nonetheless, some papers have been published that make an attempt. For instance, Letchford et al. [11] use bootstrapped standard errors with big search data. Other papers like Rashad [14] address it through careful and theoretically grounded limitations to specific searches. Our results show the challenges of doing this right. For instance, bootstrapped errors (as drawn by Letchford) are unlikely to retain the spatial autocorrelation in their data, but spatially correlated data have quite high spurious correlations. While some scholars attempt to use a variety of machine learning methods to select models [1,28], others have been deeply skeptical [20,29] of this attempt.

Quantifying spurious correlations across a range of distributions in big data in the ways this paper illustrates is thus crucial for several reasons. It can help scholars assess the extent of spurious correlation in the data at hand. This, in turn, highlights the critical importance of best practices concerning variable selection when using big search data. Theoretically driven reasons for the selection of search terms are crucial. Additionally, the full disclosure of the range of terms is also considered but discarded in any model designed for forecasting or scientific investigation, along with out-of-sample prediction.

The remainder of this report proceeds as follows. Section 2 briefly describes the methods used to generate random variables, with which spurious correlations were identified using Google Correlate (more details are in the Supplementary Material). Section 3 summarizes the results of the investigation. Section 4 discusses the implications and concludes.

## 2. Materials and Methods

This section discusses the approach taken to generate spurious correlations and the random numbers that were the basis for these spurious correlations. The primary analysis of this paper concerns correlations between random numbers and big search data, and this is the main focus of the methods section. The secondary analysis involves examining spurious correlations between pairs of random numbers.

The primary analysis examines the association between real search data and simulated variables generated using a random number generator. To analyze the association with big search data, for each of the six probability density functions, we drew approximately 500 random variables. We then sought correlations between each of these random variables and search frequency variables identified by Google Correlate. This approach guarantees that all correlations are entirely spurious. Because the random-number-simulated cross-state and time series variables cannot have true causal correlations with real-world data, any correlations identified must be spurious. Since the correlations are with real-world data, we can measure variability in the vulnerability to spurious correlations in the field rather than merely in simulations, as in Calude and Longo [18]. The critical identifying assumption for our analysis is that because the random-number-simulated target variable, with which correlations in search data are identified was generated randomly, any observed correlations with the big search data will necessarily be entirely spurious.

Unlike Vigen [21], where the possibility of an indirect or spurious causal process remains (i.e., pet and alcohol spending may not cause each other, but some third factor such as loneliness may partly influence both), the correlations we examine in this study reflect no causal relation to any real-world data. The correlations we examine are thus *entirely* spurious [26]. Calude and Longo [18] also examine simulated correlations that are entirely spurious, but their study does not examine correlations where one of the variables is drawn from real big search data.

We selected the following four widely modeled distributions to generate simulated cross-sectional data for the US states: the uniform distribution, the normal distribution, the gamma distribution, and a spatially correlated distribution. The following two common functions were used to generate simulated time series data: a mean-reverting random variable composed of the average of a series of random normal draws of mean zero and a simple

Gaussian random walk. Code and samples, as well as data, are available in the replication archive for this paper, along with a Supplementary Material that provides a more detailed description of the random variable generation process (see the data statement below).

In brief, the cross-sectional random variables were generated as follows. The uniform, normal, and gamma distributions were all modeled through a series of independent draws from the respective distributions, one draw per state. Approximately 500 random variables were generated for each probability distribution. The spatially correlated distribution was generated as follows. First, each state was assigned a random draw from the uniform distribution. Then values of neighboring states were averaged, and a new state value was created from a weighted average of each state's value (multiplied by 0.8) with the average value of the neighboring states (multiplied by 0.2). The process of averaging neighbors and computing a weighted average was repeated. Because it does leverage information on the geographic location of states, one might argue that the spatially correlated distribution does contain some information that is not random. On the other hand, most real-world comparator datasets are likely also to have some type of spatial correlation.

The U.S. time series random variables were generated as follows. For the random walk variables, each weekly value of the time series is the prior weekly value plus a new draw from the standard normal distribution. In the mean-reverting normal distribution, each weekly value represents the sum of 53 random normal draws. In the transition from week $t$ to week $t + 1$, the 'earliest' of these 53 draws is dropped, and a new draw is added.

As noted already, for each of these probability distributions, we created a set of approximately 500 random variables. These variables were then uploaded using the "Enter your own data" option in the Google Correlate platform [27]. Figure 1 illustrates the basic input window from Google Correlate. The tool allowed users to enter a search term and find terms with highly correlated search frequencies. It also allowed users to enter their own frequency variable using the "Enter your own data" link shown at the right. We used this method to upload our randomly generated variables and to find search terms that were highly correlated with these random variables.

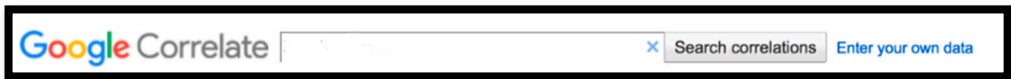

**Figure 1.** Screenshot of Google Correlate input page.

Google Correlate allowed users to identify search frequency data that correlated highly (above 0.6) with each of the randomly generated variables. The 0.6 limitations is imposed by the blockade on serving up correlations below 0.6 in Google Correlate. Presumably, this censoring represented a judgment call by someone at Google that correlations below 0.6 were likely to be spurious. Unfortunately, as we discuss below, many entirely spurious correlations well above 0.6 were nonetheless observed in our study. Typical output from Google Correlate is displayed in Figures 2 and 3, and Table 1 below.

Figures 2 and 3 illustrate the data, and the Google Correlate output using two examples in the native Google Correlate interface of completely spurious correlations between our random variables and search frequency data that was identified by Google Correlate. In Figure 2, a randomly generated cross-state variable drawn from the gamma (1, 1) distribution is correlated at 0.82 with searches for "jordan webb". The figure shows the cross-state frequency of each variable. In Figure 3, a random walk time series is overlaid on the search frequency graph for "liking". In each case, the correlation is reported between the simulated random variable (e.g., "Gamma1" and "RandWalkWK9") and the web search frequency of the search term identified as highly correlated with that variable by Google Correlate. Table 1 below provides examples of the full set of search terms provided by Google Correlate in response to some of the random variables we uploaded.

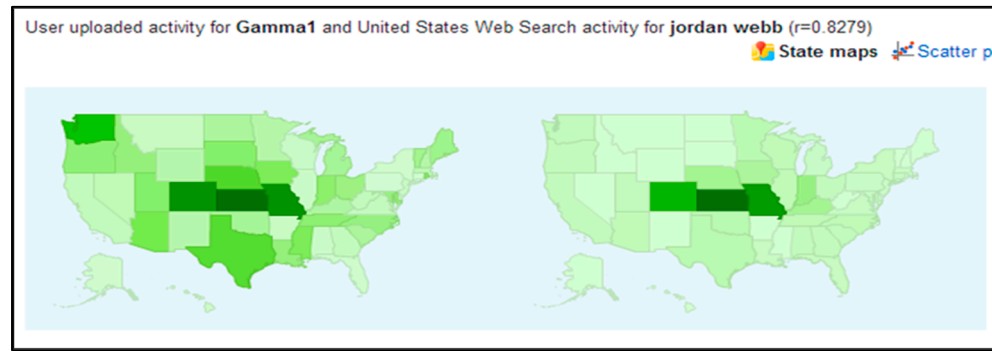

**Figure 2.** Screenshot of identified spurious correlation from Google Correlate. Gamma1 is a randomly generated cross-state distribution resulting from draws of the gamma (1, 1) distribution.

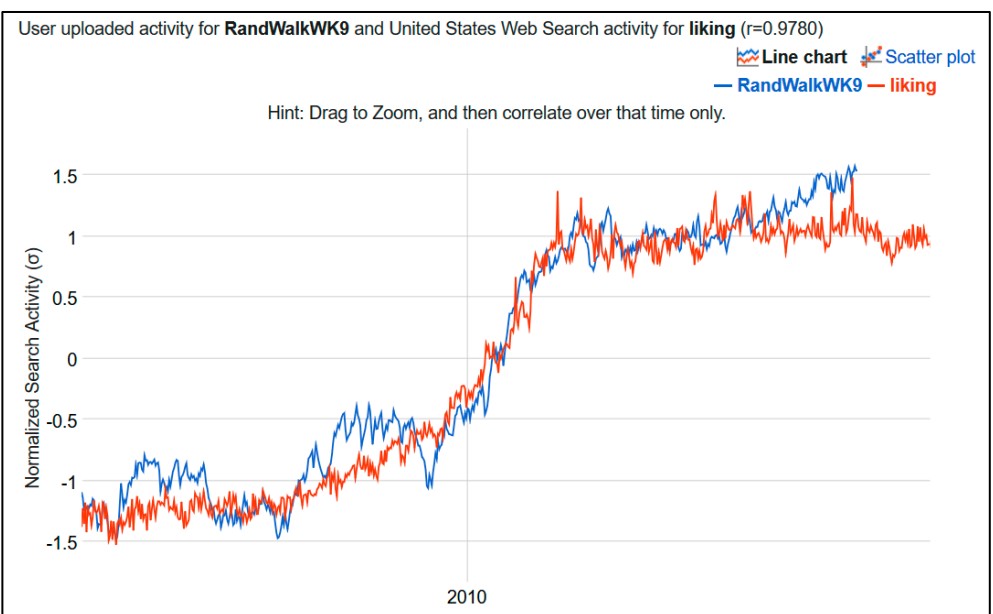

**Figure 3.** Screenshot of identified spurious correlation from Google Correlate. RandWalkWK9 is a random walk variable generated by adding successive standard normal draws to generate a weekly time series. The time frame for the weekly time series is from January 2004 through January 2016.

The total number of correlated terms identified by Google Correlate varied and, in some instances, was censored by the Google Correlate system. Not all these correlations were collected for every run, as collection involved time-consuming scraping of many pages of results, and only the top 90 results were available to be recorded. For example, Table 1 lists the 90 most correlated terms from a gamma distribution experiment and a random walk experiment. It is worth noting that clear similarities can be detected among some of the terms, while other terms seem completely unrelated. This likely reflects the depth of the spurious correlation problem since all were highly correlated with the random-number-generated variable. For instance, the random walk appears to match many terms about pregnancy. However, then it also picks up high associations with a range of other terms. The minimum correlation with that random walk list of 90 terms was 0.9353. That means that these terms were very highly spuriously correlated with the randomly generated variable. There were, presumably, hundreds or thousands of other search terms with nearly as high an association with this random walk that were excluded by the fact that Google Correlate only allowed access to the top 90 most correlated terms.

**Table 1.** Examples of 90 most correlated search terms from sample gamma and random walk runs.

Gamma Run (top 90 results with maximum correlation 0.72 to minimum correlation 0.65): whistling, ron jones, red ticking, purdy, james alan, auburn golf, city of mount vernon, maximilien, weather mount vernon, eastgate park, tucker park, pine box, richard pope, nancy stewart, auburn theater, liquid lime, rock orchestra, state abortion laws, hunter tree, elma,, amazon grocery, burger master, state adoption, foley library, diagnoser, stanley and, lynnwood apartments, state congress, college running, baker lab, motor trucks, state polls, mount vernon zip, the rainier, scan tv, callison, hope place, ivan the gorilla, hooverville, auburn medical center, weight loss for life, pignataro, funtasia, ballard, gates hall, days inn auburn, elma, pi, weather in mount vernon, ken hutcherson, 5 tv, genealogy search engine, state congressional districts, state rivers, 1077, ups university, capital mall, mill creek, phinney, idiot pilot, lakewood cinema, center laser, narrows bridge, white center, the airlock, emerald ridge, bainbridge high, bainbridge high school, avacyn restored spoilers, healthfinder, small works, the mural, state pta, the other coast, the patriarchs, mount vernon police department, row to hoe, home lodge, bonney, evergreen medical center, treehouse for kids, bellevue high school, three dollar bill, james g, reid realty, the family pet, figgy, bellevue high, teneriffe, egg nest.

Random Walk Run (top 90 results with maximum correlation 0.9532 to minimum correlation 0.9353): inmate locator, chase, period calculator, best wordpress, 26 weeks pregnant, 14 weeks pregnant, 29 weeks pregnant, view text messages, jail inmates, 15 weeks pregnant, nyc midtown, 18 weeks pregnant, wordpress page, landers mclarty, chances of getting pregnant, 33 weeks pregnant, pain during pregnancy, hard to get, on a mac, wordpress admin, mucus in stool, weeks pregnant, miami dade inmate, franklin tn, 32 weeks pregnant, clip in hair, how to text, madison heights mi, email to text, 25 weeks pregnant, do girls like, skype history, 33165, clip in hair extensions, find my cell, songs like, what song goes, could i be pregnant, what is a good, like a guy, lansing mi, macbook pro screen, 33186, m and t, find my cell phone, best pdf, 23 weeks pregnant, clip in, plugin for wordpress, dade inmate search, your high, canton mi, fg, 19 weeks pregnant, allen tx, girl you like, miami dade inmate search, okemos mi, gluten free?, uitableview, what does te, acoustic chords, county jail inmates, chase bank in, birds barbershop, in charlotte nc, chico ca, what is the easiest, pregnant symptoms, xps to pdf, altamonte springs fl, dream mean, during pregnancy, good name, how far along am i, 31 weeks pregnant, how far along am, lls, livonia mi, chase on, restore to factory, a pregnancy test, move wordpress, in memphis tn, artists like, how do you tell, grand rapids mi, jquery scroll, kp.org, frederick md.

In addition to the primary analysis described above involving spurious correlations between big search data and random number variables, we also examine the baseline spurious correlations between pairs of our random number variables—the association between the randomly generated variables themselves. To analyze spurious associations between random variables, we drew one million pairs of random variables for each pairing of probability distributions and recorded the resulting correlations. This provides a baseline for comparison with the results of the Google Correlate spurious correlations. To facilitate this comparison, the number of observations was kept the same as for the weekly time-series (n = 630) and states-cross-sectional (n = 50) datasets. In correlations between the cross-sectional and time-series variables, only the first 50 observations of the time series were used.

## 3. Results

Table 2 reports the maximum spurious correlation achieved for paired draws of two random variables for each combination of the time series and cross-sectional variables. In total, 1,000,000 pairs of each type of variable were created using the procedures described above, and the correlations between the variables were recorded. This provides a baseline concerning typical maximum spurious correlations likely to be associated with different types of random variable draws. For the mathematical derivation of probabilities for selected distributions, see [30].

**Table 2.** Maximum spurious correlations in simulations testing 1,000,000 pairs of random variables following specified distributions.

|  | Uniform | Normal | Gamma (1, 1) | Spatial | Random Walk | Mean Reverting |
|---|---|---|---|---|---|---|
| Uniform | 0.66 |  |  |  |  |  |
| Normal | 0.62 | 0.63 |  |  |  |  |
| Gamma (1, 1) | 0.59 | 0.61 | 0.80 |  |  |  |
| Spatial | 0.68 | 0.62 | 0.62 | 0.73 |  |  |
| Random Walk | 0.59 | 0.64 | 0.62 | 0.59 | 0.98 |  |
| Mean Reverting | 0.61 | 0.63 | 0.66 | 0.61 | 0.88 | 0.82 |
| Max Overall | 0.68 | 0.64 | 0.80 | 0.73 | 0.98 | 0.88 |

The results in Table 2 indicate that for the random variables we examine, quite high spurious correlations occur when one million pairs of variables are examined, though some variables generate larger spurious correlations than others. The normal distribution random variable has the lowest maximum spurious correlation across the runs at 0.64, while the random walk has the highest at 0.98.

Table 3 turns to our main analysis as follows: spurious correlations of random variables with big search data variables, as identified using Google Correlate. The table summarizes the Google Correlate results for all the random variables created from each probability distribution. Overall, large spurious correlations are frequent. At least one spurious correlation larger than 0.6 was identified 22–99% of the time for the variables following each of the probability distributions. While the uniform and standard normal distributions had relatively small frequencies of such correlations (less than 1/3), the gamma and random walk had the highest.

**Table 3.** Frequency of spurious correlations identified by Google Correlate by distribution of random variable.

| Probability Distribution from Which Random Variables Were Drawn | N of RVs | Portion of Random Variables with Spurious Correlation > 0.6 | Mean Largest Correlation per Variable (Standard Deviation) | 95th Percentile of Largest Spurious Correlation across Variables | Largest Correlation Found across Variables |
|---|---|---|---|---|---|
| Spatial | 500 | 68% | 0.66 (0.04) | 0.72 | 0.78 |
| Gamma (1, 1) | 600 | 97% | 0.71 (0.06) | 0.82 | 0.91 |
| Std. Normal | 499 | 33% | 0.63 (0.02) | 0.66 | 0.71 |
| Uniform | 500 | 22% | 0.63 (0.02) | 0.64 | 0.7 |
| Mean-Reverting | 500 | 76% | 0.69 (0.06) | 0.78 | 0.85 |
| Random Walk | 500 | 99% | 0.87 (0.08) | 0.97 | 0.98 |

Note: For each probability distribution, random variables were drawn, and the correlations between those variables and big search data were sought using Google Correlate. Observations are censored if correlation < 0.6, which biases the mean largest correlation but not the 95th percentile or maximum estimates.

The table also identifies the 95th centile of the largest correlation identified, which is arguably relevant for thinking about what level of correlation would be considered strong enough to be unlikely to be spurious in big search data at $p < 0.05$. See [30] for a discussion of these issues and some analytical results focussed on the normal distribution. The lowest value is for the uniform distribution at 0.64, and the highest is for the random walk at 0.97.

The probability density function, which generated the most spurious correlations, was the Gaussian random walk time series. At least one spurious correlation above 0.60 was identified 99 percent of the time for random variables following this distribution. On average, the largest spurious correlation identified by Google Correlate for a given random walk variable was 0.87. The highest single spurious correlation identified across our experiments was also identified for this distribution—a correlation of 0.98 between a randomly generated random walk variable and a real search frequency variable. As previously noted, the 95th percentile for these estimates (taking account of the missingness)

is 0.97. This result suggests that for a time series believed to be following a random walk, even a correlation of 0.96 is likely not large enough to reject the null hypothesis that the correlation is spurious at the $p < 0.05$ level. The result is also consistent with our simulations reported in Table 2, as follows: the random walk had the highest identified spurious correlation between pairs of random variables in the simulations as well.

The random variables with the next-largest frequency of spurious correlations above 0.6 were the ones drawn from the cross-sectional gamma (1, 1) distribution. Nearly every randomly generated variable following the gamma distribution had at least one spurious correlation identified above 0.6 (97%). On average, the largest spurious correlation identified with a given random variable was 0.71. The largest spurious correlation identified across all 600 random variables generated that followed this distribution was 0.91. The 95th percentile for these estimates (taking account of the missingness) is 0.82. This result suggests that for a cross-sectional variable believed to be following a gamma (1, 1) distribution, even a correlation of 0.81 is likely not large enough to reject the null hypothesis that the correlation is spurious. We have not investigated other gamma distributions, so we cannot comment on whether this high frequency is specific to gamma (1, 1) or applies to the entire family of distributions.

The random variables following a uniform distribution had the smallest frequency of high spurious correlations. Google Correlate identified a spurious correlation above 0.6 only 22 percent of the time. Additionally, when spurious correlations were identified, they were typically relatively small (on average 0.626). The largest correlation identified was 0.70. However, it is worth emphasizing that even for the uniform distribution, large (above 0.6) spurious correlations were hardly absent—they appeared more than one-fifth of the time. This result is largely consistent with the expectations one would drive from Table 2, except that Table 2 might potentially lead one to expect the least spurious correlations to be identified for the normal distribution. Perhaps this result can be explained as follows: the uniform distribution has a higher maximum correlation with itself than the normal distribution does with any distribution, but the normal distribution has a higher maximum correlation with all of the other distributions than the uniform distribution does.

Figure 4 provides histograms and density plots for the maximum spurious correlation identified per random variable, conditioned upon observation of a correlation above 0.6. Whereas the time series random walk probability distribution generates consistently very high spurious correlations, the cross-sectional gamma (1, 1) random variable typically finds spurious correlations that are lower but substantial. The distributions appear skewed, likely because of ceiling effects and censoring. All distributions are censored at a correlation of 0.6 because Google Correlate did not report results if no correlations were found above 0.6. Thus, for some distributions, a substantial portion of the observations in the uncensored distribution would have been below 0.6 (e.g., for the spatially correlated distribution, 32 percent of observations were censored). On the other hand, the random walk distribution appears to be encountering a ceiling effect, with few censored observations below 0.6 (1%) but many observations close to the theoretical maximum of one.

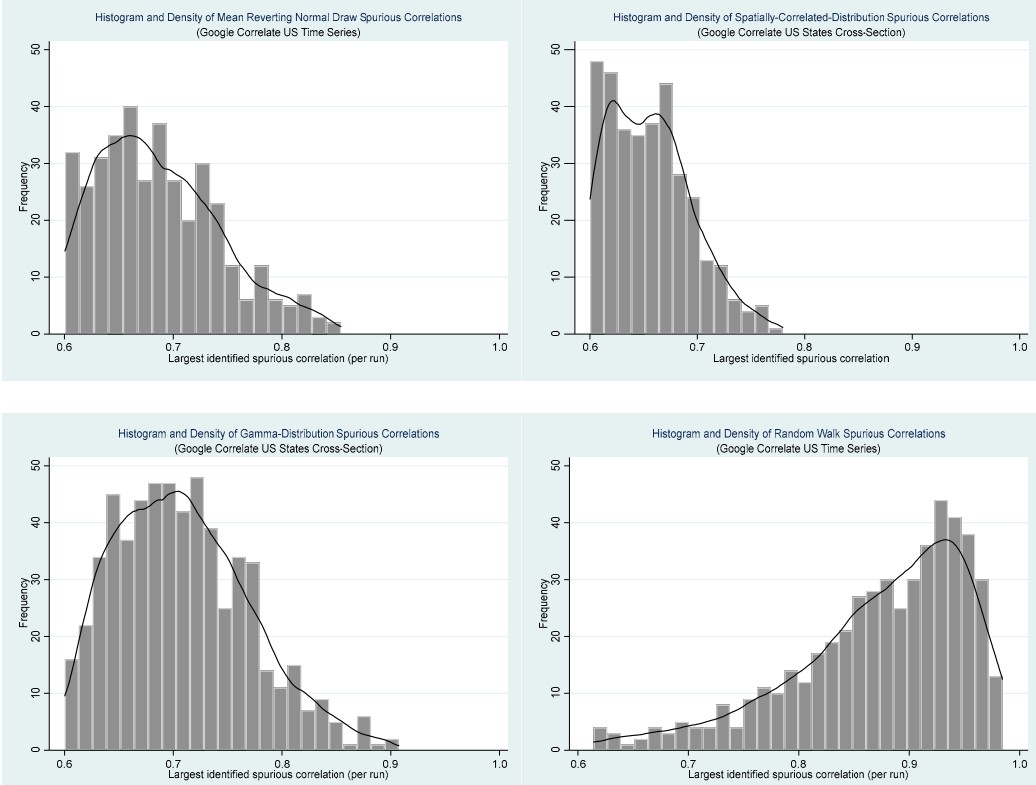

**Figure 4.** Density of maximum spurious correlations: gamma (1, 1), spatially correlated, Gaussian random walk, and mean-reverting normal distributions. Note: each distribution is censored at r = 0.60, which excludes some experiments from the figure (see Table 3 for the portion of experimental runs that were censored).

## 4. Discussion

This study is the first to quantify the magnitude of the spurious correlation problem for big search data. Consistent with previous theoretical arguments [19], we find that the magnitude of the problem is very large. At the extreme, the Gaussian random-walk random variables we simulated had, on average, a spurious correlation of 0.87 with a search term, and 99 percent of random walk variables had at least one spurious correlation above 0.6. The results in this study also imply that the severity of spurious correlation risk varies in ways that the previous literature has not fully appreciated. The most 'dangerous' variables to try to predict (high risk of spurious correlations) are those that follow probability distributions such as the gamma (1, 1) distribution and the Gaussian random walk. The lowest risk among the probability distributions examined is for the uniform and normal distributions, but even for these distributions, we frequently found spurious correlations above 0.6 using Google Correlate.

The key lesson is that forecasters, analysts, and reviewers must emphasize study design choices that can reduce the risk of spurious correlation. Studies that are not transparent about the total number of candidate predictor variables examined and rejected should be the most suspect as follows: did the authors reject many (millions) candidate variables? If we believe the results of such a study, we are likely to be fooling ourselves. As noted earlier, the Bonferroni correction with $\alpha = 0.05$ and ten million hypotheses tested is $0.05/10,000,000 = 0.000000005$. Studies that indiscriminately mine for highly correlated terms are likely to encounter enormous difficulties [18–20,29], although efforts continue to identify approaches that will work [1,30]. While the specific tool used to collect the data used in this study (Google Correlate) has since been discontinued by Google, other tools continue to offer the capacity to examine many searches [31].

A study with a pre-registered research plan in which a small number of search terms were identified based on theory or observation before data collection can perhaps mitigate the spurious correlation risk. Some examples of studies that appear to have met these criteria include Johnson et al. [10], who studied search term choice by the target population to validate the selection of candidate terms. Other studies [3,14,15] examine a limited number of terms or types of searches directly connected to a specific target phenomenon based on theory or involve replication with new data of patterns identified in previous studies [7]. However, even the better studies are often less explicit than they should be about whether any other candidate search terms were examined and discarded.

The risk of spurious correlation varies in systematic ways. Therefore, it is also vital to model the correct distribution in simulation studies that attempt to construct adjusted confidence intervals by accounting for the probability of spurious correlation, such as [11]. Where the probability distribution of the phenomena of interest can be simulated accurately, a series of random draws from that distribution can potentially provide a distribution of spurious correlations to facilitate appropriate null hypothesis tests. The thresholds will, at times, be extremely high. Large spurious correlations of 0.87 are roughly what one would expect on average for a Gaussian random walk probability distribution in big search data. However, for other distributions, such as the uniform and normal cross-sectional, such high correlations are clearly less common (95 percent of Google Correlate spurious correlations are below 0.66 for the normal distribution, and 95 percent are below 0.64 for the uniform distribution). The right distribution, including key characteristics such as the extent of spatial and autocorrelation, must be modelled correctly in any effort to bootstrap confidence intervals.

It is important to acknowledge the limits of the analysis reported here. We examine only six probability distributions and only big search data. It is possible that higher (or lower) levels of spurious correlation would be found for other distributions or other types of big data. For instance, while we examine the gamma (1, 1) distribution, a gamma distribution with different alpha or beta parameters might have generated fewer (or more) spurious correlations. Similarly, a non-Gaussian random walk might have generated different levels of spurious correlation. The results are also limited to big search data of the sort accessible through Google Correlate. It is possible that lower (or higher) levels of spurious correlation might be identified within other big fat datasets or that the relative levels of spurious correlation identified across different random variable probability distributions might vary in those contexts. The investigation of such possibilities is left for future work.

The magnitude of the spurious correlation risk means that analysts ought to approach any study mining big search data to generate predictive variables with extreme caution. Our results quantify the risk of finding "scientific fool's gold" [32] in the form of large spurious correlations and show it varies substantially across probability distributions for big search data. Given that some time-series can have average maximum spurious correlations above 0.8 in Google Correlate, any effort at "enforcing very tight significance levels to avoid an excess of 'false positives'" due to spurious correlation for such series as in [28] must set the significance level very high indeed.

Quantifying the risk of spurious correlations in the big data being used, as this study is the first to do for big search data, is an important step toward making appropriate inferences and accurate forecasts. Quantifying these spurious correlations and their likely magnitude for various distributions has value for several reasons. First, analysts can make progress toward accurate inference. Second, they can avoid unwarranted credulity. Third, they can demand appropriate disclosure from the study authors.

**Supplementary Materials:** The following supporting information can be downloaded at: https://www.mdpi.com/article/10.3390/forecast5010015/s1. References [33–37] are cited in the Supplementary Materials.

**Author Contributions:** Conceptualization, J.T.R.; methodology, J.T.R. and R.J.R.; software, J.T.R. and R.J.R.; validation, J.T.R. and R.J.R.; formal analysis, J.T.R. and R.J.R.; data curation, R.J.R.; writing— original draft preparation, J.T.R.; writing—review and editing, J.T.R. and R.J.R.; visualization, J.T.R. All authors have read and agreed to the published version of the manuscript.

**Funding:** This research received no external funding.

**Data Availability Statement:** Data and other replication materials are available at "Replication Data for: Measuring and Answering the Challenge of Spurious Correlations in Big Search Data", https://doi.org/10.7910/DVN/UW1UYR (accessed on 26 December 2022), Harvard Dataverse.

**Conflicts of Interest:** The authors declare no conflict of interest.

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
