# Peer review of "Assessing Spurious Correlations in Big Search Data"

_forecasting, doi:10.3390/forecast5010015_

Round 1

Reviewer 1 Report (Previous Reviewer 2)

My previous suggestions were reflected in the current manuscript and I have no further comments. The authors have now added a notice that the Google Correlate software used for the analysis is no longer available. I think this caveat is very important and should have been included in the manuscript from the very beginning.

Author Response

Overall comments

We are both very grateful for the thorough and careful analysis of our paper provided by the reviewers.  The paper has benefited greatly from their thoughtful suggestions and comments. In the memo below, we outline the ways we have addressed each of their comments.

Response to Reviewer 1. 

Reviewer 1 wrote:

My previous suggestions were reflected in the current manuscript and I have no further comments. The authors have now added a notice that the Google Correlate software used for the analysis is no longer available. I think this caveat is very important and should have been included in the manuscript from the very beginning.”

We apologize for not having this included in the manuscript at the beginning.  We didn’t know it until we were prompted to check into this issue by the reviewer’s questions. 

Please see the attached document for responses to all reviewers.

Reviewer 2 Report (New Reviewer)

Manuscript Number forecasting-2153999:

Assessing Spurious Correlations in Big Search Data

The work attempts to quantify the risk in the context of big search data to find spurious correlations arise at especially high frequencies for variables having probabilistic distributions. Although it is important this development, it needs strong improvements in order to be published. I have the following comments about the paper:

Major Comments

  1. The contribution of the paper is limited, where the authors didn't show the mathematical formulae of the distributions used in the manuscript (as this distribution can have different parameterizations. Also, if they can write the mathematical definition of the uniform, normal, gamma spatially auto-correlated distributions will be better.
  2. They should provide MSE of the simulated results for each of the distributions used.
  3. What are the algorithms followed to generate the random numbers. Please write the down the procedures step by step.
  4. Do you used normal distribution or standard normal distribution to generate the random numbers? Please clarify.
  5. Please discuss on the main results of your work. For instance, the distribution that provide large spurious correlations at high frequencies among the competing distributions.
  6. Add more recent and relevant references.

Minor Comments

  1. Please revise your abstract
  2.  Line 15 “Among other things, it highlights how essential it is that analysts disclose 15 the number of potential predictors that were considered (and discarded)”.
  3. Please recast lines 52-61. In particular, remove (which) in line 54.
  4.  Line 97, random number simulated cross-state and time series…
  5. You used (modelled in line 111 and modeled in line 118). Be consistent please.  
  6. Please add legend and remove scatter plot in Figure 1(a).
  7. Line 213, please add Mean Reverting Normal.
  8. The conclusion should be improved in lines of the obtained results.
  9. Please check the language of the paper.

Author Response

Overall comments

We are both very grateful for the thorough and careful analysis of our paper provided by the reviewers.  The paper has benefited greatly from their thoughtful suggestions and comments. In the memo below, we outline the ways we have addressed each of their comments.

Please see the attached document for our responses to all reviewers.

Response to Reviewer 2.

Reviewer 2 wrote:

“The work attempts to quantify the risk in the context of big search data to find spurious correlations arise at especially high frequencies for variables having probabilistic distributions. Although it is important this development, it needs strong improvements in order to be published. I have the following comments about the paper:”

We appreciate the reviewer’s comments about the paper, and we outline below the ways we have sought to address them.  We hope that the reviewer will agree with us that the process of response to these revisions has made the work stronger, and we are very grateful for the careful read of the paper provided by this reviewer.

Reviewer 2 continued:

Major Comments

  1. “The contribution of the paper is limited, where the authors didn't show the mathematical formulae of the distributions used in the manuscript (as this distribution can have different parameterizations. Also, if they can write the mathematical definition of the uniform, normal, gamma spatially auto-correlated distributions will be better.

  1. “What are the algorithms followed to generate the random numbers. Please write the down the procedures step by step.
  2. “Do you used normal distribution or standard normal distribution to generate the random numbers? Please clarify.”

Reviewers 2, 3 and 4 made comments about wanting to see a more explicit and extensive development of the process for creating the random variables.  We have responded by expanding the discussion in the text slightly, and by adding a detailed appendix (this could be published with the paper, or simply made available in the replication archive at the editor’s discretion).  The appendix details the process by which each of the random variables used in our experiments were created.  It clarifies issues like whether the normal distribution or standard normal distribution was used (and explains why it doesn’t ultimately matter) and develops step-by-step the procedures used to create the variables and import them into Google Correlate.

We determined that explicating the process of drawing each random variable in detail in the paper would require too much space, so we have added an appendix which explains in more depth and step by step the process for generating the simulated data. At the option of the editor and reviewers, we are happy to have the appendix either appear with the printed paper, or be produced only in the data archive. https://doi.org/10.7910/DVN/UW1UYR. Or a shortened version could be integrated into the paper itself.  Our preference would be to have the appendix printed in the paper.

“2. They should provide MSE of the simulated results for each of the distributions used.”

This comment led us to think more deeply about the range of summary measures we could include in Table 3 (formerly Table 1 as we also reordered the tables). We have added standard deviations to the column summarizing average maximum spurious correlation, and have also added a 95th percentile measure which shows the magnitude of spurious correlation achieved by 5 percent of our spurious correlations for each of the random variables, a useful quantity potentially for thinking about inference, and also a useful metric because it is not biased by the censoring at correlation 0.6.  We are grateful that this comment from Reviewer 2 led us to explore ways to more extensively analyze our data, and we believe that the result is a much stronger paper with clearer implications.

  1. “Please discuss on the main results of your work. For instance, the distribution that provide large spurious correlations at high frequencies among the competing distributions.”

We have expanded our discussion of the results in several places.  There are now several paragraphs discussing in detail the results summarized in Table 3.  We also added additional more detailed discussion in the conclusion, and have revised the abstract to make the key take-aways clearer.

  1. “Add more recent and relevant references.”

We have added several additional recent references to the paper with a focus on studies tackling the challenge of spurious correlations in big data.  We have also noted more explicitly that some of the studies we cite are foundational articles (e.g. the Varian piece) which have been cited by thousands of later papers, many of which are also pursuing analyses of big search data.

“Minor Comments

  1. Please revise your abstract
  2.  Line 15 “Among other things, it highlights how essential it is that analysts disclose 15 the number of potential predictors that were considered (and discarded)”.
  3. Please recast lines 52-61. In particular, remove (which) in line 54.
  4.  Line 97, random number simulated cross-state and time series…
  5. You used (modelled in line 111 and modeled in line 118). Be consistent please.  
  6. Please add legend and remove scatter plot in Figure 1(a).
  7. Line 213, please add Mean Reverting Normal.
  8. The conclusion should be improved in lines of the obtained results.
  9. Please check the language of the paper.”

We have attempted to address all of these comments and suggestions, including careful re-reads by both authors to check the language of the paper, revising the figures, correcting the specific typos and problems identified, and reworking the conclusion.

Reviewer 3 Report (New Reviewer)

The manuscript is written rather carelessly. Though the authors are residing in a native English speaking country (it seems so!), their grammar in the manuscript is far from being correct. Some sentences are too long and they are hard for the reader to make sense. Perhaps they can shorten the sentences by breaking them into twos and threes. There are many places where inappropriate uses of words or similar cases appear, e.g., correlation between data” rather than “correlation among variables”. I advise the authors to make those correction so that text is easier to understand than the current one. For me, certainly the text is not in the publishable form. Some editing service may be used to improve the grammar and text of the manuscript.  

Define and explain Big Search Data.  Use Big Search Data instead of Big Data wherever the case. 

“Google Flu” should not be a key word.

Explain “Vigen’s vignettes”.

Add little more information on “Google Correlate” and how the authors have collected data through it.

Add more on quantification and finding spurious correlation in Big Data, etc.

Can the authors give some explanation or reasons for their observation in the Lines 74 to 79.

Have the authors checked potential correlations in their simulated data. If so, what are they? Is their target variable (Line 102) a simulated one?

It is advisable that the steps described in Lines 110 through 140 are shown as an algorithm using pseudo-codes. It can include the steps doe with Google Correlate. Simulated data should be clearly explained

In Table 1, what is “Maximum cross draws”? Shouldn’t it be an integer? Why is it larger than the third column of the Table. Revise the Table 1. 

All graphs in Figure 2 should be shown using same x-axis range of [0.6, 1] for ease of comparison. Can the authors give reasons why some of the graphs are right-skewed and others are left-skewed? Add a note on frequency of finding correlations less than 0.6

Move the paragraph from Line 215 to Line 226 to the section “Introduction” and expand it by adding some more information

Re-write or update the section “Discussion” so that authors contribution is clear. Now it is rather hazy.

Author Response

Overall comments

We are both very grateful for the thorough and careful analysis of our paper provided by the reviewers.  The paper has benefited greatly from their thoughtful suggestions and comments. In the memo below, we outline the ways we have addressed each of their comments.

Please see the attached document for our responses to all reviewers.

Response to Reviewer 3.

Reviewer 3 wrote:

“The manuscript is written rather carelessly. Though the authors are residing in a native English speaking country (it seems so!), their grammar in the manuscript is far from being correct. Some sentences are too long and they are hard for the reader to make sense. Perhaps they can shorten the sentences by breaking them into twos and threes. There are many places where inappropriate uses of words or similar cases appear, e.g., correlation between data” rather than “correlation among variables”. I advise the authors to make those correction so that text is easier to understand than the current one. For me, certainly the text is not in the publishable form. Some editing service may be used to improve the grammar and text of the manuscript.”

We have attempted to address these concerns about the language of the article: each one of us read and re-read the article looking for sentences to break up, and other ways to improve the prose.   

Reviewer 3 continued:

“Define and explain Big Search Data.  Use Big Search Data instead of Big Data wherever the case. 

“Google Flu” should not be a key word.”

We have removed “Google Flu” as a key word.  We have added more definition of big search data and big data, and have evaluated each instance in which we use the words “Big Data” to evaluate whether “big search data” or “big data” would be more appropriate.

Reviewer 3 continued:

“Explain “Vigen’s vignettes”.”

We have added significant additional explication of Vigen’s work, including recharacterizing what Vigen produced as “data cartoons” a term that perhaps better captures the almost exclusively visual nature of his presentation.

Reviewer 3 continued:

“Add little more information on “Google Correlate” and how the authors have collected data through it.”

We have added more information about Google Correlate and additional discussion of how data was collected. Some of this discussion appears in the text of the paper, where we added an additional figure and more explanation.  Other information appears in the appendix.

Reviewer 3 continued:

“Add more on quantification and finding spurious correlation in Big Data, etc.”

We have expanded our discussion of these issues and contributions in the abstract, the conclusion, and the discussion.

Reviewer 3 continued:

“Can the authors give some explanation or reasons for their observation in the Lines 74 to 79.”

We have added more explanation.

Reviewer 3 continued:

“Have the authors checked potential correlations in their simulated data. If so, what are they? Is their target variable (Line 102) a simulated one?”

We have revised this text to clarify that it is indeed simulated.

Reviewer 3 continued:

“It is advisable that the steps described in Lines 110 through 140 are shown as an algorithm using pseudo-codes. It can include the steps doe with Google Correlate. Simulated data should be clearly explained”

As discussed above in our response to reviewer 2, we determined that explicating the process of drawing each random variable in detail in the paper would require too much space, so we have added an appendix which explains in more depth and step by step the process for generating the simulated data. At the option of the editor and reviewers, we are happy to have the appendix either appear with the printed paper, or be produced only in the data archive. https://doi.org/10.7910/DVN/UW1UYR

Reviewer 3 continued:

“In Table 1, what is “Maximum cross draws”? Shouldn’t it be an integer? Why is it larger than the third column of the Table. Revise the Table 1.”

We have revised the headings and other aspects of the table (now renumbered as Table 3) in order to make these quantities clearer. More discussion of the range of ways we have attempted to improve this table is found in our response to Reviewer 2 above. We also added several paragraphs that explain table results in depth for particular probability distributions.  We believe this additional explanation, in combination with our efforts to clarify the table, should make the table much clearer for readers, and we are very grateful for the reviewer’s suggestions concerning the table.   

Reviewer 3 continued:

“All graphs in Figure 2 should be shown using same x-axis range of [0.6, 1] for ease of comparison. Can the authors give reasons why some of the graphs are right-skewed and others are left-skewed? Add a note on frequency of finding correlations less than 0.6”

We have revised the axis scale to make it consistent, and have added discussion of the sources of skew in the various graphs.

Reviewer 3 continued: “Move the paragraph from Line 215 to Line 226 to the section “Introduction” and expand it by adding some more information”

We have moved this section up to the introduction of the study and have expanded it as suggested.  This was an excellent suggestion from the reviewer: this paragraph fits much better in the introduction.

Reviewer 3 continued: “Re-write or update the section “Discussion” so that authors contribution is clear. Now it is rather hazy.”

We have rewritten this section in line with the excellent suggestions from reviewers 2 and 3, and believe that we have succeeded in making the contributions clearer.  We remain open to additional insights from the reviewers about ways to further refine these sections.

Reviewer 4 Report (New Reviewer)

This study shows how spurious correlations arise in the big internet search data that follow certain distributions. The paper claims this fact via a real data case study by choosing few distributions.

The major concerns of this study are:

1) Authors claim that: "spurious correlations arise at especially high frequencies for variables following gamma and spatially auto-correlated distributions, and random walks" would require proving that all/most variables that follow gamma and spatially autocorrelated distributions are spurious. However, in this study authors first simulate data from certain distributions with fixed parameters and find out some spuriously correlated variables that correlate well with target simulated data. These two findings are not consistent. For example,

     a) there could be other variables that are gamma distributed but not spuriously correlated.

     b) there could be variables that are gamma distributed but not spuriously correlated.

     c) there could be other variables that are gamma distributed with different parameters and not spuriously correlated.

   Due to all such reasons, it is not easy to establish such a claim based on one case study. 

2) This is related to 1). Authors don't provide any sensitivity study by varying the parameters or datasets.  

3) Is there any scientific or theoretical reasoning for the authors' claim mentioned in 1)?

4) It is not surprising that for any big data, the likelihood of spurious correlation increases. What would be the added benefit of quantifying the extent of spurious correlation if the spuriously correlated are likely to be discarded?  

Other particular comments are as follows:

1) Line 111-140: The data generating mechanism section should constitute equations and clear explanations of each term and the reasonings for a specific set of parameter values. It is hard to follow the methods section without that.

  2) Figure 1: What is the x-axis time window? There is no label except for 2010.

3) Line 231: What does it mean by bound the appropriate confidence bound? 

Author Response

Overall comments

We are both very grateful for the thorough and careful analysis of our paper provided by the reviewers.  The paper has benefited greatly from their thoughtful suggestions and comments. In the memo below, we outline the ways we have addressed each of their comments.

Please see the attached document for our responses to all reviewers.

Reviewer 4

Reviewer 4 wrote:

“This study shows how spurious correlations arise in the big internet search data that follow certain distributions. The paper claims this fact via a real data case study by choosing few distributions.

The major concerns of this study are:

1) Authors claim that: "spurious correlations arise at especially high frequencies for variables following gamma and spatially auto-correlated distributions, and random walks" would require proving that all/most variables that follow gamma and spatially autocorrelated distributions are spurious. However, in this study authors first simulate data from certain distributions with fixed parameters and find out some spuriously correlated variables that correlate well with target simulated data. These two findings are not consistent. For example,

  1.    a) there could be other variables that are gamma distributed but not spuriously correlated.
  2.    b) there could be variables that are gamma distributed but not spuriously correlated.
  3.    c) there could be other variables that are gamma distributed with different parameters and not spuriously correlated.

   Due to all such reasons, it is not easy to establish such a claim based on one case study.”

We have revised our discussion in the paper to make clearer the limitations in terms of these possibilities.  For instance, the line in the abstract which the reviewer quoted has been modified to make clearer that the claims are restricted to the specific instantiations of the distributions we simulated in the context of our data analysis.  We agree to some extent with the point the reviewer made in bullet c that some other Gamma distribution might have different levels of spurious correlation, and we have clarified in the discussion and reporting of results to make this limitation of the analysis clearer, including a new paragraph specifically addressing this limitation in the conclusion, and comments in new paragraphs that were added to the results section.

However, we believe that running random variables through Google Correlate does a pretty good job of establishing answers with regards to points a and b above.  As summarized in Table 3, there are indeed some random variables following the Gamma (1,1) distribution which do not have high spurious (above 0.6) correlations that could be identified by Google Correlate.  Approximately 3 percent of Gamma(1,1) random variables fall into this category based upon our analysis. The study acknowledges this possibility, and indeed reports on it.  It is one of our results that they exist.  What we show is that there are such variables, but big search data is big enough that there simply do not seem to be many of them.  

We are grateful that reviewer 4 brought these issues to our attention, as they have helped us clarify our analysis and discussion in important ways.

Reviewer 4 then continued: 

“2) This is related to 1). Authors don't provide any sensitivity study by varying the parameters or datasets.”

The use of multiple probability density functions to model our random variables was intended on our part to be an exploration of the sensitivity of our results to various probability distributions that might be followed by random variables.  But the point is well taken in the sense that one could always conceive of having done more. We now much more explicitly acknowledge this limitation.  We hope that despite this limitation the reviewer will agree with us that an analysis of six different random variables does provide enough information to make a contribution, and we are grateful that this critique has helped us to more appropriately clarify the limitations of our study.

“3) Is there any scientific or theoretical reasoning for the authors' claim mentioned in 1)?”

Most analyses that have aimed to calculate the scale of spurious correlations (e.g. the 2018 paper by Jianqing Fan. Qi-Man Shao. Wen-Xin Zhou. "Are discoveries spurious? Distributions of maximum spurious correlations and their applications." Do not explicitly examine multiple distributions. To address the gap the reviewer pointed out in this comment, we have added additional analysis that compares the results from the Google Correlate analysis with results from analysis of pairs of random variables. This appears in the new Table 2.  This helps clarify that the patterns identified for the Google Correlate analysis in terms of maximum correlations found largely parallel the results of simply comparing correlation magnitudes between random variables.  We have also taken additional care to ensure that the discussion of the results is more appropriately couched in terms of the limits of the empirical analyses we conducted. These results are we believe important empirical contributions of the paper.  In brief, we don’t think that this is something one would have known would be the case in big search data without empirical investigation, although the simulations in Table 2 provide a foundation for expectations that are largely borne out in the empirical analysis of Google Correlate results. 

Reviewer 4 continued:

“4) It is not surprising that for any big data, the likelihood of spurious correlation increases. What would be the added benefit of quantifying the extent of spurious correlation if the spuriously correlated are likely to be discarded?”

We are not sure if we fully understand the reviewer’s point in the final part of this item “if the spuriously correlated are likely to be discarded.” Indeed, the concern of various critics of poorly constructed big data analyses is that the spurious correlations are unfortunately likely to be hard to detect and hence some may fail to be discarded.

The reviewer’s broader point appears to be about whether quantifying spurious correlations is valuable. We have sought to sharpen our discussion of this point. We believe that quantifying these spurious correlations and their likely magnitude for various distributions has value for several reasons. First, analysts can make progress toward accurate inference. Second, they can avoid unwarranted credulity. Third, they can demand appropriate disclosure from study authors.

In addition, we have expanded our discussion in the paper to highlight what we believe is one of the potential (though more difficult to achieve) possible benefits of quantification.  Table 3 now contains an additional column of data reporting the 95th percentile of spurious correlations identified.  That is, spurious correlations this high or larger only occur 5 percent of the time.  We believe this helps highlight the advantages of quantification: arguably, if the distribution of the variable being predicted was known, such estimates could potentially allow for testing of null hypotheses based upon the empirical likelihood of a data mining process identifying spurious correlations of a specific magnitude.    

Reviewer 4 continued:

“Other particular comments are as follows:

“1) Line 111-140: The data generating mechanism section should constitute equations and clear explanations of each term and the reasonings for a specific set of parameter values. It is hard to follow the methods section without that.”

A similar point was made by reviewers 2 and 3. As discussed above in our response to reviewers 2 and 3, we determined that explicating the process of drawing each random variable in detail in the paper would require too much space, so we have added an appendix which explains in more depth and step by step the process for generating the simulated data. At the option of the editor and reviewers, we are happy to have the appendix either appear with the printed paper, or be produced only in the data archive. https://doi.org/10.7910/DVN/UW1UYR. Or a shortened version could be integrated into the paper itself. 

“ 2) Figure 1: What is the x-axis time window? There is no label except for 2010”

This is the time window and information provided by Google Correlate. The time frame is from January 2004 through January 2016 because this is the time frame we used for all of our time series random variables.  We have added information to the figure caption to provide more detail.  And we appreciate the reviewer bringing the vagueness of the graphic to our attention.

“3) Line 231: What does it mean by bound the appropriate confidence bound?”

We have expanded our discussion of this point and replaced this language with what we hope is clearer discussion.  We are grateful to the reviewer for bringing this unclear wording to our attention. 

Round 2

Reviewer 2 Report (New Reviewer)

The authors have put effort to revised the manuscript as suggested. The paper is now recommended for publication.

This manuscript is a resubmission of an earlier submission. The following is a list of the peer review reports and author responses from that submission.

Round 1

Reviewer 1 Report

Title: Assessing Spurious Correlations in Big Search Data

Authors: Jesse Richman, Ryan Roberts

Xiao-Li Meng’s 2018 article in the Annals of Applied Statistics, using more general statistical theory, accomplishes what this article seeks to do by simulation in specific discrete portions of the parameter space.  I’m afraid if the topic the authors raise came up, I would suggest someone go read Meng’s article instead of this paper.  To be fair, the simulation setup is different than Meng's theory but the message is basically the same.

This paper is certainly of pedagogical value and will give readers a feel for the issues.  Perhaps it could be submitted to an educational journal or used for other purposes. I’m not sure I see the case for a scientific contribution however.

One specific suggestion, if the authors continue down the path they are on: I appreciate the authors’ decision to randomize a variable, but real variables in real datasets tend to be substantially more correlated than that.  The whole notion of a ‘dataset’ presupposes that the variables contained in it have something to do with one another after all.  Conceivably the authors could estimate from a large number of real datasets the empirically reasonably levels of correlation and then build out their simulation to be more realistic.

Author Response

The authors are very grateful for the thoughtful comments offered by both of the reviewers.  As we detail below, we believe that responding to these comments offered us an opportunity of make this paper stronger in a number of different ways.  In the memo below, we discuss how we responded to each of the comments and suggestions made by the reviewers.  The comments from reviewers are indented below, followed by a discussion of our responses and revisions.

Reviewer 1

>>Xiao-Li Meng’s 2018 article in the Annals of Applied Statistics, using more general statistical theory, accomplishes what this article seeks to do by simulation in specific discrete portions of the parameter space.  I’m afraid if the topic the authors raise came up, I would suggest someone go read Meng’s article instead of this paper.  To be fair, the simulation setup is different than Meng's theory but the message is basically the same.

We appreciate the suggestion that we examine the analysis in Meng’s article and we have taken this as a starting point for a discussion (a new paragraph in the discussion section) which uses the Meng article to put what this paper is about in the context of the broader range of challenges raised by big data for statistics and statistical inference. We highlight that the Meng paper is principally focused on the problem of inference from big but potentially low quality samples of a variable, while our focus is principally on forecasting, causal analysis, and prediction based upon correlations between putative predictor variables and some dependent variable of interest.  These are clearly closely related challenges, and the work by Meng points to some of the directions for future development, as we now discuss.

>>This paper is certainly of pedagogical value and will give readers a feel for the issues.  Perhaps it could be submitted to an educational journal or used for other purposes. I’m not sure I see the case for a scientific contribution however.

We appreciate the comment about pedagogical value.  We hope that this paper does indeed provide such value by giving readers a “feel for the issues” and also highlighting to reviewers that at minimum they must demand disclosure from authors about any alternative measures of variables that were considered.  We have added a sentence emphasizing this point to the abstract and in various other areas aimed to strengthen our discussion which we hope will help ensure that the contribution we are aiming to make to the quality of scientific analyses and the knowledge about the potential for spurious correlation is not missed.

>>One specific suggestion, if the authors continue down the path they are on: I appreciate the authors’ decision to randomize a variable, but real variables in real datasets tend to be substantially more correlated than that.  The whole notion of a ‘dataset’ presupposes that the variables contained in it have something to do with one another after all.  Conceivably the authors could estimate from a large number of real datasets the empirically reasonably levels of correlation and then build out their simulation to be more realistic.

We have added some discussion of this possibility to the discussion section of the paper as we think this could be valuable future work.  We think that even without an examination of such correlations, however, there is value in studying the kind of completely spurious correlations that we examine here.  Completely spurious correlations are admittedly an extreme case, but we believe that they have pedagogical and scientific value.  A very large number of papers have been published over the last decade using search data.  For example, Varian’s paper “Predicting the Present” has been cited 3112 times, many of them citations by studies that are similarly using search data for forecasting purposes.  This is potentially very useful data, and clearly an area worth examining because of its wide application and potential for both scientific progress and abuse.  We hope that future work can expand the scope of the analysis to other types of data as well.

Reviewer 2 Report

The topic of spurious correlations in big data is very important. The current contribution delas with a specific source of big data - search data.

My main comment is on the design of the conducted research. Table 1 presents results of an experiment, where high correlations (>0.6) were recorded. It is clear that for each distribtion 500 (or 600) draws were generated. But it is not clear how many real world variables were correlated to them! This should be clarified. Otherwise, the results are unclear. E.g. how to interpret 68% at the fist line of Table1? Does it mean that in 0.68*500 =340 daws at least one correlation higher than 0.6 was found? But how many correlations were computed for that single instance? 

My second comment is on Google Correlate (lines 122-128). Since I am not familiar with this tool, I tried to find out how it works. It would be useful to provide some information on how it can be accessed and used.

Minor remark: line 102: it is not clear what "spatially correlated distribution" means. Please add here a note that the distribution is described in next paragraphs.

Minor remark: are the codes already available somewere? I was not able to find them.

Author Response

The authors are very grateful for the thoughtful comments offered by both of the reviewers.  As we detail below, we believe that responding to these comments offered us an opportunity of make this paper stronger in a number of different ways.  In the memo below, we discuss how we responded to each of the comments and suggestions made by the reviewers.  The comments from reviewers are indented below, followed by a discussion of our responses and revisions.

Reviewer 2

>>The topic of spurious correlations in big data is very important. The current contribution delas with a specific source of big data - search data.

We agree with the reviewer that the topic of spurious correlation in big data is very important.

>>My main comment is on the design of the conducted research. Table 1 presents results of an experiment, where high correlations (>0.6) were recorded. It is clear that for each distribtion 500 (or 600) draws were generated. But it is not clear how many real world variables were correlated to them! This should be clarified. Otherwise, the results are unclear. E.g. how to interpret 68% at the fist line of Table1? Does it mean that in 0.68*500 =340 daws at least one correlation higher than 0.6 was found? But how many correlations were computed for that single instance? 

>>We have added some text to clarify the table, and to explain that the Google Correlate interface was designed to limit the total number of correlations identified to 90, so the total number is hard to quantify.  We also added a new table which illustrates the results by providing the top 90 spurious correlations with a couple of our variables. 

>>My second comment is on Google Correlate (lines 122-128). Since I am not familiar with this tool, I tried to find out how it works. It would be useful to provide some information on how it can be accessed and used.

We added a reference to a Google white paper which explains the Google Correlate tool.

>>Minor remark: line 102: it is not clear what "spatially correlated distribution" means. Please add here a note that the distribution is described in next paragraphs.

We have implemented this suggestion.

>>Minor remark: are the codes already available somewere? I was not able to find them.

We have published the codes in the dataverse that is linked in the data disclosure statement at the bottom of the article.  The doi link should now take a reader to this replication archive.

Round 2

Reviewer 2 Report

Explanation of Table 1 is clear now.

However, more information about Google Correlate tool should be provided. The added reference is a bit outdated (2011). Is the tool still available? I was not able to run it. Please provide a clear up-to-date link to the tool. I was even not able to follow links from the provided excel files.

Author Response

Reviewer 2 made two points, and we will respond to each in turn.  The review is quoted, followed by our responses. 

“Explanation of Table 1 is clear now.”

We are grateful that the table is now clear, and we appreciate the suggestions made by the reviewer which helped us improve our presentation.

“However, more information about Google Correlate tool should be provided. The added reference is a bit outdated (2011). Is the tool still available? I was not able to run it. Please provide a clear up-to-date link to the tool. I was even not able to follow links from the provided excel files.”

We are very grateful for this additional follow-up query from Reviewer 2 concerning Google Correlate.  Through some additional investigation, we have now become aware that after we collected the data for this study, Google decided to shut down the Google Correlate tool.  

We have now added a paragraph to the introduction and a sentence and a reference in the discussion section.  In these places in the article, we now note the shut down of Google Correlate, and direct readers to the Google Trends tool and other tools which continue to allow scholars to identify the search frequencies of each search term and continue to provide functionality related to similar searches.

While we have had fun playing with it, we believe that the shutdown of the Google Correlate tool is in some sense wonderful news.  As our study makes clear, it was far too easy to find extremely high completely spurious correlations using the tool, and the shutdown of this tool means that researchers will now have to work at least a little bit harder to find the sorts of massive numbers of spurious correlations we explore in this study.  

As we have reflected on the situation, we believe that the end of Google correlate makes publication of a study such as this one even more important. This study is now the main opportunity for the scientific community to learn from data collected using the Google Correlate system.  As reviewer 1 noted in the first round of review, the Google Correlate data allows the paper to make a vital pedagogical and methodological point about the prevalence of spurious correlations in search and similar big data.  While search data remains just as massive, and just as full of potential spurious correlations, the window Google Correlate provided into this massive potential for spuriousness has now closed. Thus, this study likely took advantage of the last opportunity to use the Google Correlate system to study spurious correlations in big search data.

Once again, we are very appreciative of the time and effort put into the review of this article by both of the reviewers, and we stand ready to address as well as we possibly can any additional concerns or questions.

The revised paper is attached below. 
